# Preclinical Activities of Epigallocatechin Gallate in Signaling Pathways in Cancer

**DOI:** 10.3390/molecules25030467

**Published:** 2020-01-22

**Authors:** Mehdi Sharifi-Rad, Raffaele Pezzani, Marco Redaelli, Maira Zorzan, Muhammad Imran, Anees Ahmed Khalil, Bahare Salehi, Farukh Sharopov, William C. Cho, Javad Sharifi-Rad

**Affiliations:** 1Department of Medical Parasitology, Kerman University of Medical Sciences, Kerman 7616913555, Iran; mehdi_sharifirad@yahoo.com; 2Endocrinology Unit, Department of Medicine (DIMED), University of Padova, via Ospedale 105, 35128 Padova, Italy; maira.zorzan@vimset.it; 3AIROB, Associazione Italiana per la Ricerca Oncologica di Base, 35046 Padova, Italy; marcoredaelli@email.it; 4Venetian Institute for Molecular Science and Experimental Technologies, VIMSET, Pz. Milani 4, Liettoli di Campolongo Maggiore (VE), 30010 Venice, Italy; 5University Institute of Diet and Nutritional Sciences, Faculty of Allied Health Sciences, The University of Lahore, Lahore 54590, Pakistan; mic_1661@yahoo.com (M.I.); aneesahmedkhalil@gmail.com (A.A.K.); 6Student Research Committee, School of Medicine, Bam University of Medical Sciences, Bam 44340847, Iran; 7Department of Pharmaceutical Technology, Avicenna Tajik State Medical University, Rudaki 139, Dushanbe 734003, Tajikistan; 8Department of Clinical Oncology, Queen Elizabeth Hospital, Hong Kong, China; 9Phytochemistry Research Center, Shahid Beheshti University of Medical Sciences, Tehran 1991953381, Iran

**Keywords:** epigallocatechin-3-gallate, EGCG, catechin, green tea, cancer preventive, pharmacological activities

## Abstract

Epigallocatechin gallate (EGCG) is the main bioactive component of catechins predominantly present in various types of tea. EGCG is well known for a wide spectrum of biological activities as an anti-oxidative, anti-inflammatory, and anti-tumor agent. The effect of EGCG on cell death mechanisms via the induction of apoptosis, necrosis, and autophagy has been documented. Moreover, its anti-proliferative action has been demonstrated in many cancer cell lines. It was also involved in the modulation of cyclooxygenase-2, oxidative stress and inflammation of different cellular processes. EGCG has been reported as a promising agent target for plasma membrane proteins, such as epidermal growth factor receptor. In addition, it has been demonstrated a mechanism of action relying on the inhibition of ERK1/2, p38 MAPK, NF-κB, and vascular endothelial growth factor. Furthermore, EGCG and its derivatives were used in proteasome inhibition and they were involved in epigenetic mechanisms. In summary, EGCG is the most predominant and bioactive constituent of tea and may play a role in cancer prevention.

## 1. Introduction

Tea has been known to be the world most renowned non-alcoholic beverage since ancient times and it is currently being utilized by two-thirds population of the world, owing to its taste, stimulating effects, unique aroma, and associated health perspectives [1,2]. Various types of tea like green, black and oolong tea are derived from the *Camellia sinensis* (L.) plant containing variety of components among which polyphenols are most significant ones. Basic differences among all of these teas depends upon the stage of fermentation processes from which they are produced. Purposely, green tea is not fermented while black tea and oolong tea are completely and partially fermented, respectively. Amongst all of the investigated teas around the globe, green tea is well studied, owing to its health promoting benefits. In general, tea based phenolics possess protective action against numerous metabolic syndromes.

Several nutraceutical aspects of green tea extracts depends upon the concentration of phenolics and their associated derivatives. The major biologically active moiety present in leaves of green tea are classified as catechins that nearly account for 25–35% on dry weight basis. This catechin group comprises eight phenolic flavonoid constituents, specifically, catechin, epicatechin, gallocatechin, epigallocatechin, catechin gallate, epicatechin gallate, gallocatechin gallate, and epigallocatechin gallate (EGCG) [3,4]. Among above mentioned polyphenols, EGCG is the most vital tea based catechin which is considered to be the main reason for bioactivity of green tea [5,6,7,8].

Catechins are plant secondary metabolites [9,10]. They possess numerous functions in plant survival, growth, and metabolism, but they can also interact with other living organisms if ingested or came into direct contact. Catechins or flavanols probably constitute the most abundant subclass of flavonoids and they are essentially represented by (–)-epigallocatechin-3-gallate (EGCG), (–)-epigallocatechin (EGC), (–)-epicatechin-3-gallate (ECG), and (–)-epicatechin (EC) [11]. These last four bioactive compounds are found in large amount in green tea (leaves of *Camellia sinensis*) and cocoa/chocolate, but they are also present in many fruits or seeds, such apples, blueberries (*Vaccinium myrtillus*) and grapes (red wine and beer) and peanuts [12]. This review focuses on catechins that are derived from green tea and in particular on EGCG, which represents 50% to 80% of all *Camellia sinensis* catechins (200–300 mg/brewed cup of green tea) [13]. Moreover, we analyzed preclinical works examining its pharmacological and biomolecular mechanism of action.

Epigallocatechin gallate is the most bioactive catechin that is predominantly present in tea. Among all of the tea types, it is found at maximum comncentration in green tea leaves. EGCG molecule (Figure 1) comprises two aromatic structures that are co-joined by three carbon bridge structure (C6-C3-C6) along with hydroxyl group (OH) at simultaneous carbons i.e., 3’, 4’, 5’ of ring-B. Carbon-3 of ring-C is esterified with a gallate molecule (features of catechins originated from tea are responsible for bioactivity due to position and number of OH-group on the rings. They regulate their capability to interact with biological matter via hydrogen bonding, a hydrogen and electron transferring process contained by their antioxidative potentials. The presence and absence of a galloyl molecule differentiates EGCG from remaining three catechins [14].

Pure epigallocatechin gallate is classified as an odourless crystal and/or powder available in white, pink, or cream colour. It is considered to be soluble in water as a colorless and clear solution (5 mg mL^−1^). It is also soluble in methanol, tetrahydrofuran, acetone, pyridine, and ethanol. EGCG have a melting point at 218 °C [15].

Even though EGCG is found to be the most predominant and bioactive constituent present in tea, it is considered to be poorly stable in aqueous solutions and poorly soluble in oils and fats [16,17,18,19]. This poor stability and solubility restricts its direct addition in food products. Numerous delivery systems are, in practice, to preserve its structural integrity and shield EGCG from degradation [18,20,21,22,23,24]. Additionally, structural modification in EGCG has also aided in elevating its lipophilicity [25,26].

Numerous physical aspects like pH, light, oxidants contents, oxygen, temperature, and concentration of EGCG, influence the stability of epigallocatechin gallate [17,26,27,28,29]. Epimerization and auto-oxidation are two most important reactions that result in instability of EGCG and loss of its health promoting properties [30]. EGCG is oxidized at low concentration and temperature i.e., 20 to 100 μM & < 44 °C, respectively. On the other hand, it is epimerized to gallocatechin gallate at elevated temperature (greater than 44 °C) and acidic conditions (pH 2 to 5.5) [30,31,32,33]. EGCG at a concentration of 300 mg per litre has been found to be stable at 25 °C (24 h) in a pH range of 3 to 6, nonetheless if kept at 4 °C & 25 °C for 24 h (pH 7), a loss of 25 & 83%, respectively, was observed [29]. 

EGCG is an abundant source of phenolic OH-groups that leads to its health escalating properties. Various scientific studies (in vivo and in vitro) have been carried out to authenticate the potential of EGCG as an anti-oxidative, antiinflammatory, and anti-cancerous agent [34,35].

Wei and co-authors have reported that EGCG exerts an anti-tumor effect through the inhibition of key enzymes that participate in the glycolytic pathway and the suppression of glucose metabolism [36]. Shang and coworkers proposed that tea catechins in combination with anticancer drugs are being evaluated as a new cancer treatment strategy [37]. It has a potential role as adjuvant in cancer therapy and it could enhance the effect of conventional cancer therapies through additive or synergistic effects, as well as through the amelioration of deleterious side effects [38]. EGCG interacts with catalase and highlighted as an anticancer drug [39]. Green tea polyphenols are reported as a translational perspective of chemoprevention and treatment for cancer with concrete influence on signaling pathways [2]. EGCG as a powerful antioxidant, antiangiogenic, and antitumor agent, and as a modulator of tumor cell has the potential to impact in variety of human diseases [40].

This review was prepared systematically searching for articles, papers, books from different sources, such as Embase-Elsevier, Google Scholar, Ovid, PubMed, Science Direct, Scopus, and Web of Science. The search strategy was based on the combination of different keywords, such as EGCG, cancer, tumor, mechanism of action, preclinical, and clinical data, while considering any time of publication. Only sources written in English were included.

## 2. Occurrence of Epigallocatechin-3-Gallate in Different Foods

The close relationship between food and disease is well known [41], thus EGCG consumed in food could be a natural potential source of health benefits, even in complex and intricated disorder as cancer.

Principal catechin in tea is EGCG, which is also known to be responsible for various health modulating perspectives tht are related to utilization of tea. According to Wu et al., (2012), a cup of green tea prepared by adding 2500 mg leaves in 0.2 L of water contains 90 mg of EGCG [42]. The content of EGCG in black tea is less when compared to green tea, mainly due to polymerization of catechins [43]. EGCG that are derived from tea possess various bioactive properties and, therefore, have attained attention as potent functional and nutraceutical ingredient in food and pharmaceutical industry. Table 1 lists EGCG contents in some selected foods.

## 3. Signaling Pathways Involving EGCG: Proliferation, Differentiation, Apoptosis, Inflammation, Angiogenesis and Metastasis

A plethora of biomolecular mechanisms has been demonstrated and proposed to be involved in the use of this catechin in preclinical models. These can include, but are not limited to, the induction of apoptosis, cell cycle change, antioxidant activity, modulation of drug metabolism enzymes, etc. EGCG possesses multiple pharmacological activities, depending on cancer cell type, thus reflecting different protein expression and epigenetic mechanisms. Every cell model can express and modulate at convenience its genetic background, and this is at the basis of the diverse biomolecular targets triggered by EGCG. More consistently, EGCG has been analyzed in different tumor cell lines and less frequently in animal models: this paragraph investigates the association of EGCG with proliferation, differentiation, apoptosis, inflammation, angiogenesis, and metastasis, i.e., the main signaling pathways and processes of living cells (Figure 2).

### 3.1. Cell Death Induction (Apoptosis, Necrosis, Autophagy)

The effects of EGCG on cell death induction, including apoptosis and autophagy, have been investigated in several preclinical studies. Jian and colleagues demonstrated that EGCG induced apoptosis of human liver hepatocellular carcinoma HepG2 cells and rat pheochromocytoma PC12 cells by decreasing the Bcl-2 protein and up-regulating Bax, when used at >20 μM [51]. The proposed mechanism of action of EGCG was related to Human antigen R (HuR, also known as ELAVL1), which can control amyloid protein precursor (APP) via HuR-Erk1/2-APP pathway. APP is implicated in Alzheimer’s disease and cancer, blocking ATP with EGCG might be useful in the treatment.

A combination of clofarabine (640 nM and 50 nM in MCF7 and MDA-MB-231 cells, respectively) and EGCG (10 μM) also induced apoptosis in human breast cancer cells, followed by hypomethylation of the tumor suppressor gene RARβ (retinoic acid receptor β) and the upregulation of RARβ, PTEN, and CDKN1A genes [52]. Clofarfabine can inhibit DNA methylation reaction and, thus, its combination with EGCG can have a role in blocking breast cancer development.

The effects of EGCG (>80 μM) on tongue squamous cell carcinoma were recently investigated, showing that it reduced proliferation while promoting apoptosis in CAL27 and SCC 15 cells via the Hippo-TAZ signaling pathway [53]. Similar to other tumor cell models, EGCG (10–200 µM) induced apoptosis in a dose-dependent manner in human thyroid carcinoma cell lines by increasing the Bax/Bcl-2 ratio, thus pointing to an increase of mitochondrial-mediated apoptosis and the involvement of EGFR/RAS/RAF/MEK/ERK signaling pathway [54]. In addition to apoptosis, cell treated with EGCG has been associated with autophagy regulation. The treatment of human embryonic kidney cell line (HEK293T cells) with EGCG increased cell viability by promoting autophagy through mTOR pathway [55]. In hepatocellular carcinoma cells, EGCG induced autophagy via direct interaction with LC3-I protein and, consequently, led to the degradation of α-fetal protein (AFP); an effect that suggests a potential therapeutic application for the treatment of this type of tumor [56]. EGCG-induced autophagy is also involved in the regulation of ectopic lipid accumulation in vascular endothelial cells, which indicates a potential beneficial effect in preventing cardiovascular diseases [57]. Thus, EGCG is involved in autophagy, even in different tumor cell lines, but with concentration dependent on cell tissue origin. The effect of EGCG on cell death mechanisms is well documented. EGCG has been also proposed as protective agent against necrosis induction because of its demonstrated action on reduction of oxidative stress and inflammatory responses [58].

### 3.2. Cell Cycle Arrest (Cyclin, CDK)

The effect of EGCG on cell proliferation and cycle has been investigated in various cancers. EGCG in a early work has been demonstrated to play an effective role as an anti-proliferative and chemopreventive agent breast cancer, colon cancer, and skin cancers, such as melanoma in experiments while using MCF-7, HT-29, and UACC-375 cell lines [59]. Specific effects on cell cycle modulation were also investigated in a liver cancer model in which EGCG inhibited the proliferation of liver cancer cells in G2 phase and induced apoptosis after the blocking of the cell cycle progression in the G1 phase [60]. Similar effects were supported in a study on skin cancer, in which the benefits of using polyphenols, such as EGCG, in the treatment and prevention of skin cancer were extensively presented [61]. The effect of EGCG treatment has been also investigated on different cancer types in which cell cycle modulation was demonstrated through the alteration of the pattern of cell cycle proteins with specific reference to CDKs. EGCG increased the expression of CDK1 proteins and inhibited the expression of Cyclin D1 and the phosphorylation of retinoblastoma proteins. Moreover, EGCG treatment has been associated to cell cycle arrest affecting upregulation/downregulation balance in which p1/p21, Kip1/p27, p16/INK4A, cyclin D1, cyclin E, CDK2, and CDK4 proteins are involved. This effect was demonstrated to be the first step of the apoptosis cascade induction via ROS and subsequent caspase-3 and caspase-9 action [62]. These conclusions were then supported by additional in vitro and in vivo experiments in melanoma cancer cells. EGCG treatment of melanoma cells results in the modulation of cyclin-dependent kinases inhibitor network and Bcl2 family proteins with a consequent inhibition of viability and growth, cell cycle blocking, and apoptosis induction [63]. Nevertheless, it was also demonstrated that EGCG can enhance the therapeutic effect if used in synergy with IFN in both in vitro and in vivo models of melanoma, which could be an important advantage in reducing the concentration of IFN and the toxicity of the treatment [64]. The reliability of the treatment or the synergic use of EGCG is additionally supported by a large number of in vitro and in vivo scientific reports on the effects of EGCG on cell cycle modulation and apoptosis induction without affecting normal cells [65]. The studies that are presented in this paragraph point to the usefulness of EGCG in treatment against malignancies by cell cycle modulation and cell death induction, both as therapeutic agent and synergic adjuvant [13,66].

### 3.3. COX-2

Cyclooxygenase-2 (COX-2) is involved in the modulation of different cell processes, such as oxidative stress and inflammation. COX-2 interaction was considered to be part of the mechanism since EGCG was associated with cell cycle modulation and cell death induction in different cancer models. Dysregulated activity of COX-2 were described in many cancers and pathologic conditions. EGCG was demonstrated to inhibit COX-2 both in terms of transcription and translation in human prostate carcinoma cells (androgen-sensitive LNCaP and androgen-insensitive PC-3), without interfering with COX-1 regulation, and this is an important issue in cancer research. These findings point to the use of EGCG >10 µM in the treatment or prevention of prostate cancer [67]. Furthermore, the EGCG inhibition of COX-2 was investigated on a colon cancer model, in which a significant decrease of COX-2 activity in the cells of the colon of rats treated with green tea extract pointing to a prevention in the formation of preneoplastic foci [68]. The authors fed the animals with freshly prepared an aqueous solution of green tea extract, reconstituted from the lyophilisate, at a concentration of 0.02% (wt/vol) in which EGCG was at 14% dry weight. COX-2 inhibition was also studied on the potential antitumor activity of EGCG in skin cancer [69]. EGCG was administred to mice by gavage at a dose of either 20 or 50 mg/kg body weight 1 h before topical application of the tumor promoter 12-*O*-tetradecanoylphorbol-13-acetate to the shaved back. In this paper, the inhibitor mechanism of COX-2 by EGCG was described as the cascade effect of EGCG inhibition of the catalytic activity of ERK and p38 MAPK that are the upstream enzymes that regulate COX-2 expression [69]. From another point of view, the interaction between EGCG and COX-2 were investigated in arthritis. A solution of 0.2% green tea polyphenols in the water was prepared and given *ad libitum* (the sole source of drinking water for mice). Treatment with this preparation in a collagen II-induced arthritis mouse model demonstrated to reduce the pathological picture of arthritis, with decrease of TNF-α, IFN-γ, and Cox-2 especially in mice arthritic joints [70]. Thus, it was demonstrated that the treatment of human chondrocytes with EGCG at concentration >100 μM was related to the inhibition of COX-2 and iNOS activity indicating a possible involvement of this treatment in the inhibition of cartilage resorption in arthritis [71].

### 3.4. EGFR Pathway

EGFR (epidermal growth factor receptor) is a plasma membrane protein that is characterized by an intracellular region acting as tyrosine kinase and an extracellular receptor domain. The dysregulation of EGFR expression has been associated with cellular neoplastic transformation. EGCG interaction with EGFR was demonstrated and described in its mechanism. EGCG binds to a specific laminin receptor whose expression is reported in a variety of tumor cells: this connection can interfere with the activity of the kinase-phosphatase cascades [72]. The laminin receptor has been recently related to MAPK signaling in human melanoma metastasis mechanism [73]. EGCG inhibition of EGFR and HER2 was demonstrated in colon cancer cells line (HT29), in which these proteins are overexpressed in comparison with normal tissue. EGCG at 20 μg/mL resulted in cell growth reduction, cell cycle arrest in G1 phase, and apotosis induction by the activation of caspase-3 and -9, with critical impact on EGFR/ERK and EGFR/Akt signaling pathways [74]. The adjuvant effect of EGFR in breast cancer treatment was demonstrated in different cell lines (MCF-7, MCF-7TAM, and MDA-MB-231). The combined treatment of EGCG with IIF (6-OH-11-O-hydroxyphenanthrene) reduced the invasive phenotype and up-regulated the expression of molecules that are involved in the invasion process, again acting on EGFR/Akt pathway and their regulated targets, such as CD44, extracellular matrix metalloproteinases inducer (EMMPRIN), matrix metalloproteinases, and tissue inhibitor of metalloproteinases (TIMPs) [75]. Similarly EGCG activity was also investigated in MCF-7tam (a breast carcinoma cell line resistant to the chemoagent tamoxifen), where EGCG was demonstrated to reduce the resistance to Tamoxifen and block cell growth and invasion [76]. In this last work, the EGFR/ERK pathway was downregulated, with effects on matrix metalloproteinase-2 and -9, EMMPRIN. It emerges as the effects EGCG on EGFR pathway depend on cancer cell type and suggest as EGCG can have an adjuvant role in EGFR targeted therapy approach.

### 3.5. MAPK Pathway

The MAPK pathway (or extensively “Ras-Raf-MEK-ERK pathway”) is a key signaling pathway frequently dysregulated in human cancer and can control multiple cell functions, such as proliferation and growth [77]. For these reasons, different chemotherapeutic drugs have been already clinically evaluated or are currently undergoing to specific clinical trials, for example, most promising are the MEK1/2 and Raf inhibitors [78]. In mammalian cells, three major types of MAP kinases are present; ERK, p38 MAP kinase, and JNK, and they can activate c-JNK and ELK [79]. These were the main actors studied and they were associated with EGCG interaction in the MAPK pathway. A recent work studied EGCG and candesartan (anti-hypertensive drug) in a rat model of gentamicin-induced nephrotoxicity [80]. EGCG was administred at 25 mg/kg, intraperitoneally, alone or combined with the above drugs for 21 days. While renal disease revealed increased levels of NF-κB, p38-MAPK, caspase 3, TNF-α, and interleukin-1 β, the pre-treatments with EGCG and candesartan (co-treatment had the best results if compared to compounds alone) reduced such key factors, which impacted the MAPK pathway. On the same line, a work with mice of Wang et al. showed that, in animals with artificial unilateral ureteral obstruction, the intraperitoneally administration of EGCG (5 mg/kg) for 14 days inhibited phosphorylation of ERK, JNK and p38 MAPK [81]. The effect of EGCG on mastitis in rats were evaluated by injecting lipopolysaccharide (LPS) into the duct of mammary gland. EGCG was orally given at 25 or 100 mg/kg/day in the morning for 14 days. MAPKs, NFκB-p65 and hypoxia-inducible factor-1α (HIF-1α) were decreased after the treatment with EGCG at both concentrations. The results showed that EGCG suppressed MAPK related oxidative stress and inflammatory responses [82]. In another rat disease model, EGCG exhibited cardioprotective effects in myocardial ischemia/reperfusion (I/R) injuries [83]. The authors showed that EGCG 10mg/kg intravenously administered alone or with wortmannin (PI3K inhibitor) 5 min. before the onset of reperfusion reduced p38 and JNK phosphorylation while enhancing AKT and GSK-3β, but not ERK1/2. Additionally, in artificial induced hypertensive rats, ERK, p38, and JNK phosphorylation were decreased, when the rats were treated with 50 mg/kg EGCG per day, intraperitoneally for 21 days [84]. Previously in human umbilical vein endothelial cell (HUVEC) model, EGCG induced cell proliferation inhibition during ischemia/reperfusion, but, in contrast with previous work, JNK1/2 (and c-Jun) phosphorylation was increased [85]. These data could be imputable to the different model and time of incubation proposed. Also, EGCG >50 µM inhibited angiotensin II- induced phosphorylation of ERK and p38 in HUVEC cells, influencing vascular cell adhesion molecules (VCAMs) and intercellular adhesion molecules (ICAMs) expression [86,87]. In addition, EGCG impacted AKT phosphorylation and its downstream substrates Foxo1, Foxo3a, and ERK1/2. In a prostate cancer mouse model (TRAMP mice: Transgenic Adenocarcinoma Mouse Prostate), Harper and collaborators demonstrated that EGCG given *ad libitum* as 0.06% in water reduced ERK phosphorylation, together with androgen receptor (AR), insulin-like growth factor-1 (IGF-1), IGF-1 receptor (IGF-1R), COX-2, and inducible nitric oxide synthase (iNOS) [88]. The authors claimed that the EGCG dose should mimic a human consumption of six cups of green tea and that the catechin, through the decrease of MAPK signaling, could modulate inflammation biomarkers, reduce cell proliferation, and induce apoptosis. In the same mouse TRAMP model, another work showed a key role for IGF-I/IGFBP-3 signaling pathway, with the modulation of ERK [89]. However, the authors used a solution of 0.1% green tea polyphenols (in which EGCG represented 62%) to experimental animals [89]. The two works are very similar in EGCG dose, but analyzed and demonstrated the implaction of different key factors: they are probably the two sides of the same coin, but more research is needed to confirm this hyphothesis. Moreover, in a NOD mouse, a model for Sjogren’s syndrome, oral administration of 0.2% green tea extract in *ad libitum* water for 21 days (in which EGCG represented 40%) reduced the total autoantibody levels in the serum and lymphocytic infiltration of submandibular glands (a key point of Sjogren’s syndrome) [90]. The research was also expanded in normal human keratinocytes and immortalized normal human salivary acinar cells, where EGCG was able to protect the cells by specific phosphorylation of p38 MAPK [90,91,92,93], contrary to most of works in which there is a de-phosphorylation/deactivation of MAPK pathway. Again, these apparently inconsistent results should be ascribed to different models used and the different human-derived tissue.

In SKH-1 hairless mouse model, the topical application of green tea polyphenols (in which EGCG represented 18.8%) showed a decrease in UVB-induced thickness, skin edem, a and infiltration of leukocytes, processes mediated by inhibition of ERK, c-JUN, and p38, as well as NF-kB [94]. Before the animal model, the authors demonstrated the same results in cell models in which EGCG was used at 10–40 µM [92,95]. In addition, EGCG diminished the negative effects of ultraviolet irradiation in human dermal fibroblasts, reducing the phosphorylation of MAPK, JNK, p38 MAPK, and ERK1/2 [95,96]. Collagen destruction was reduced by inhibiting matrix metalloproteinases (MMP-1, MMP-8, and MMP-13). In human AC16 cardiomyocytes, EGCG was examined for alleviating cigarette smoke medium (CSM)-induced inflammation: EGCG protected from oxidative stress, inflammation, and apoptosis, modifying interleukin (IL)-8 productions via the inhibition of ERK1/2, p38 MAPK, and NF-κB pathways [97]. In the human retinal endothelial cell (HREC) line, a preclinical model for diabetic retinopathy, EGCG was able to decrease phosphorylated p38, ERK, VEGF, and the expression of inflammatory markers, such as TNF-α, IL-6, and ICAM-1 [98]. In addition, EGCG amended the negative effect of high glucose concentrations acting on the MAPK pathway. Furthermore, EGCG in chronic myeloid leukaemia cells inhibited cell growth and induced apoptosis through the inhibition of Bcr/Abl oncoprotein and regulation of its downstream signals, such as p38-MAPK and JNK (in addition to involvement of JAK2/STAT3/AKT pathway) [99]. In a different cellular context, JB6 mouse epidermal cell line, it has been shown that EGCG could inhibit AP-1(activating protein-1)-dependent transcriptional activity through the inhibition of a c-JUN, but not ERK [100]. AP-1 is a transcription factor involved that can be stimulated by cytokines, growth factors, stress, and, consequently, implicated in proliferation and apoptosis, with overlapping role to MAPK pathway. Successively, another work remarked the importance of AP-1 (and MAPK), with the inhibition of c-JUN, and also ERK by EGCG, differently from previous research [101]. Of note that a different cell line was used (human gastric AGS cells). EGCG also impacted endothelial cell activation, where the catechin could downregulate endothelial inflammation by acting on MAPK signaling through a caveolae-regulated cell mechanism [102]. Endothelial cells are fundamental in atherosclerosis: EGCG demonstrated antiatherogenic effect in reducing oxidized LDL upregulated by c-Jun phosphorylation and blocking oxidized LDL-induced endothelial apoptosis [103]. In the pancreatic alpha TC1-6 (αTC1-6) cell line, EGCG confirmed its anti-hyperglycemia and antioxidant potential [104]. Indeed, EGCG re-established glucagon secretion and reduced ROS, while inhibiting JNK and p38 and activating AKT signaling. It is of note that a higher concentration of EGCG (100 µM) was used, if compared to previous studies, and this can potentially reduce EGCG efficacy when moved to clinical trials. Moreover, in a cell model of insulin-stimulated mitogenesis (3T3-L1 preadipocytes), it has been shown that EGCG inhibited the insulin effects, switching off signaling through both insulin receptor-β, insulin receptor substrates 1 and 2 (IRS1 and IRS2) and RAF1, MEK1/2, and ERK1/2, but not JNK [105].

Interestingly, EGCG was associated with sunitinib, which is a multi-target receptor tyrosine kinase inhibitor approved for renal carcinoma and gastrointestinal stromal tumor, in MCF-7 (breast cancer) and H460 (lung cancer) mouse xenograft tumors [106]. The authors found that synergize with sunitinib due to its ability to suppress both insulin receptor substrate-1 pathway and MAPK signaling induced by sunitinib. In another breast cancer cell line T47D, EGCG induced the phosphorylation of JNK/SAPK and p38 and modulated cyclin A, cyclin B1, and cdk proteins, triggering G2 arrest [107]. In a human colon cancer cell line (HT-29), it was found that EGCG activated phospho-AKT, phospho-ERK, phospho-JNK, and phospho-p38α/γ/δ and, consequently, induced cell death, a potential mechanism for anti-cancer effect of EGCG [108]. Previously with same cell line, Chen et al. showed the activation of caspase-3 and -9, cytochrome *c*, mitochondria, c-Jun, and MAPK pathway after treatment with EGCG, with the end result of apoptosis induction [109]. In another colon cancer cell line (SW480), EGCG induced the phosphorylation of p38, which mediated the phosphorylation of EGFR at Ser1046/1047, leading to a downregulation of the EGFR/MAPK pathway [110]. The author claimed that this phosphorylation could play a pivotal role in EGFR downregulation that is induced by EGCG. Furthermore, in HCT116 colorectal carcinoma cells, EGCG triggered the phosphorylation of ERK, JNK, and p38, which, in turn, conducted to apoptosis [111]. In human lung cancer cell line PC-9, EGCG was associated with celecoxib (a cyclooxygenase-2 selective inhibitor) provoking an activation of ERK1/2 and p38 regulated by GADD153 (DNA damage-inducible 153) expression [112]. EGCG induced apoptosis and the authors suggested that GADD153 could be a new molecular target for cancer prevention in combination with EGCG. Similarly, EGCG was found to suppress cell proliferation by upregulating BTG2 (B-cell translocation gene 2) expression via MAPK pathway in oral squamous cell carcinoma (OSCC), in particular with the phosphorylation of p38, JNK, and ERK [113].

Moreover, in anaplastic thyroid cell carcinoma (ARO cells), EGCG 50 µM showed anti-proliferative and pro-apoptotic effects, being mediated by the suppression of phosphorylated EGFR, ERK1/2, JNK, and p38 and p21, caspase-3, and cleaved PARP increase and cyclin B1/CDK1 expression decline [114]. In a previous work, human hepatoma HepG2-C8 cells and human cervical squamous carcinoma HeLa cells were used to evaluate EGCG [115]. The results indicated that catechin induced the activation of ERK, JNK, and p38 in a dose- and time-dependent manner, with stimulation of apoptosis (mediated by caspase-3) and antioxidant response element (ARE). In DU-145 prostate cancer cells, EGCG associated with ibuprofen reduced >90% tumor cell growth through the activation of both MAPK and caspases [116]. Similarly, human prostate cancer androgen-unresponsive and responsive cells (DU145 and LNCaP, respectively) that were treated with EGCG showed an increase of ERK1/2 with concomitant decrease of phosphorylated AKT and the level of PI3K, again underlining the anticancer effect of the catechin [117].

### 3.6. NF-kB Pathway

NF-κB (Nuclear factor-κB) is involved in a variety of cellular mechanisms, such as death, growth, inflammation, and immune response. NF-κB mainly acts as a regulator of gene transcription in response to oxidative stress. The interaction with EGCG was described in a variety of cell models, both in vivo and in vitro [64]. NF-κB is one of the most investigated targets in cancer research and the interaction with this pathway should be considered to be a relevant topic in cancer treatment design. The inhibitory activity of EGCG on NF-κB has been demonstrated in a large number of independent studies. Studies that were performed in normal human epidermal keratinocytes in which EGCG was used at different concentration to modulate NF-κB activity, leading to a protective effect towards the negative cascade that was induced by UV radiation. This is considered the proof of the mechanism of action of green tea as a photoprotective agent [92]. Moreover EGCG has been shown to reduce significantly the gut mucosa inflammation in response to injury, thus inhibiting the production of inflammatory factors via the modulation of NF-κB expression and other involved genes [118]. The authors showed that, in ulcerative colitis (a refractory inflammatory disease of the large intestine) rat model, EGCG intraperitoneally administered at 50 mg/kg/d could reduce the severity of the disease, through the TLR4/MyD88/NF-κB signaling pathway. Moving on cancer research, EGCG was able to activate caspases in epidermoid carcinoma cells, and this has been demonstrated to play a relevant role in the inhibition of NF-κB and induction of apoptosis [119]. Nevertheless, EGCG has been demonstrated to play a potential role in human colon cancer prevention inhibiting AP-1, c-fos, cyclin D1 promoters, and NF-κB activity in four different human colon cancer cell lines, with an IC_50_ value of about 20 μg/mL [74].

### 3.7. PI3k/AKT Pathway

Numerous studies on diseases that involve PI3K/AKT signaling investigated the possible effect of green tea administration [120]. Since green tea role in asthma treatment or occurrence need to be better clarified [121,122], the effect on molecular pathways at the basis of pathological processes has been explored. Recently, a possible mechanism of action has been proposed in brochial asthma. EGCG, being administered in drinking water 0.5 mg/mL, was demonstrated to reduce inflammation and the subsequent cellular infiltration in lung tissue of ovalbumin-challenged asthmatic mouse, inhibiting EMT via PI3/AKT pathway modulation in vivo and in vitro models [123]. On the other hand, EGCG modulation on the PI3K/AKT signaling pathway was associated to lung cancer prevention and suppression. EGCG activation/inhibition of PI3K/AKT was found to be a key switch in H1299 cells growth inhibition ad apoptosis induction [124]. Another interesting research has been carried on the neuroprotective action of EGCG in a middle cerebral artery occlusion model (MCAO), acting as a modulating agent PI3K/AKT/eNOS signaling pathway [125]. The rats were treated with 20 mg/kg EGCG, injecting it with an arterial pump and the expressions of PI3K, p-AKT, and eNOS were detected by both immunohistochemistry and Western blot. Similarly, in unilateral chronic constriction injury to the sciatic nerve, EGCG role as neuroprotective agent was confirmed in rats [126]. 50 mg/kg EGCG was intraperitoneally administered daily for three days after nerve crush injury.

### 3.8. VEGF Pathway

Angiogenesis is known to play a major role in the pathogenesis of many cancer types [127]. Given its involvement in metastasis and growth of tumor cells, angiogenesis possesses a key player in vascular endothelial growth factor (VEGF, also known as VEGF-A) and it has been intensively studied as a therapeutic target [128]. Consequently, EGCG has been investigated in the VEGF pathway, discovering its capacity to modulate angiogenesis, with stimutating effects in nervous tissue and inhibiting effects in tumor tissue. For example, in a mouse model of transient middle cerebral artery occlusion (MCAO), the authors showed that EGCG injected intraperitoneally daily for seven days could stimulate angiogenesis. EGCG had a neuroprotective effect, indeed treated mice had better neurologic outcome, a decreased infarct volume, greater vascular density, with the modulation of VEGF signaling pathway and increased Nrf2 expression (fundamental role in against ischemic stroke) [129]. Similarly, hepatic tissue from mice that were treated with EGCG showed increased heme oxygenase-1 (HO-1) and decreased vascular cell adhesion molecule (VCAM)-1 expression, with an increased expression of Nrf-2 (mediated by p38) that drives the expression of HO-1, as also demonstrated in human aortic endothelial cells (HAEC) [130]. Another group used the same mouse model (MCAO) and demonstrated the involvement of the PI3K/AKT/eNOS signaling pathway with a concomitant reduction of apoptotic rate and caspase-3 in neurons, while Bcl-2 increased in the ECGC treated set (EGCG intraperitoneally injected at 20 mg/kg for 24 h) [125]. Similarly, it was studied on the effects of EGCG on murine dendritic cells (DC), with a reduction of ERK1/2, p38, JNK, and blockade of NF-κB p65 translocation [131]. EGCG was given to cells at different doses, but 100 µM was the most effective with suppression of the phenotypic and functional maturation of DC. Differently in tumor tissue EGCG acted in a opposite way. For example, in human gastric cancer SGC7901 cells (in hypoxic state) EGCG (5, 10, 20, 40 µM), reduced expressions of HIF-1α (a subunit of a heterodimeric transcription factor hypoxia-inducible factor 1-HIF1) and VEGF with a concentration-dependent effect. In addition, in SGC-7901 mouse xenografts, 1.5 EGCG mg/day intraperitoneally injected reduced VEGF, tumor microvessel density, endothelial cell proliferation, migration, and tube formation, substantially blocking tumor growth [132]. Similarly, in gastric cancer 5-fluorouracil (5-FU) resistant cells (MGC803/FU treated with 100 μg/mL and SGC7901/FU treated with 20 µM), EGCG inhibited VEGF and HIF-1 expression, reversing 5-FU resistance that is mediated by VEGF pathway [133,134]. HIF-1 is a master regulator of cell homeostatic response to hypoxia by activating the transcription of numerous genes that are implicated in growth, apoptosis, angiogenesis, and metabolism [135]. HIF-1 is strictly connected with VEGF (it is a primary target of HIF-1) and, thus, understanding the regulation of HIF-1 will offer new insight into VEGF pathway and new potential therapeutic targets. In human gastric cancer cells (AGS) EGCG >10 µM reduce dose-dependently VEGF expression and secretion mediated by signal transducer and activator of transcription 3 (Stat3) activity [136]. In colon cancer cells (Caco2, HCT116, HT29, SW480, and SW837 lines) and SW837 mouse xenograft, EGCG impacted on VEGF/VEGFR axis suppressing VEGFR2 (a primary responder to VEGF signal), together with HIF-1α, ERK, and AKT [137]. Moreover in a different human colon cancer cell line (HT29) and derived mouse xenografts, EGCG lessened VEGF expression and its promoter activity, while decreasing the tumour growth and microvessel density in mice (treated daily intraperitoneally with 1.5 mg EGCG), and modulating cell proliferation and apoptosis in HT29 cells [138]. In breast cancer cells (MCF-7), HIF-1α and VEGF was decreased by EGCG in a dose-dependent manner (25, 50, 100 mg/L) [139], similarly to non-small lung cancer cells (A549), in which the catechin (25, 50, 100 μM) was able both to inhibit HIF-1α protein expression (stimulated by IGF-I, insulin growth factor-1, a growth hormone mediating cell growth) and the formation of capillary tube-like structures on a model of extracellular matrix [140]. In head and neck squamous cell carcinoma (YCU-H891) and breast carcinoma cell lines (MDA-MB-231), EGCG blocked VEGF activity, thanks to the inhibition of constitutive activation of Stat3 and NF-κB [141]. The results showed that EGCG could modulate a plethora a key factors, such as TNF-α, Rantes, monocyte chemotactic protein-1 (MCP-1), intercellular adhesion molecule-1 (ICAM-1), nitric oxide (NO), VEGF, and MMP-2, through the inhibition of NF-κB and MAPK signaling pathways (downregulation of p-IκBα, p65, p-p65, p-p38, p-ERK1/2, and p-AKT). EGCG repressed hypoxia- and serum-induced HIF-1α and VEGF, through MAPK and PI3K/AKT pathways, in another hepatoma cell line (HepG2) and in human cervical carcinoma cell line (HeLa) [142]. In a group of acute myeloid leukemia (AML) cell lines harboring FLT3 mutation (FLT3 is a commonly mutated gene in AML), EGCG 60 µM (and (−)-epigallocatechin (EGC), (−)-epicatechin-3-gallate (ECG), and (+)-Catechin (C)) showed reduced levels of FLT3 and the suppression of phosphorylated MAPK, AKT, and STAT5 [143]. Moreover, in B-cell chronic lymphocytic leukemia (B-CLL), EGCG 3–25 µM reduced VEGF-R1 and VEGF-R2 phosphorylation (fundamental in B-CLL survival and growth) and induced apoptosis by caspase-3 activation, poly-adenosine diphosphate ribose polymerase (PARP) cleavage, and B-cell leukemia/lymphoma-2 protein (Bcl-2) [144,145].

Additionally, in non-tumoral cell lines, EGCG demonstrated a concrete role. In human microvascular endothelial cells (HMVEC) used as a model of angiogenesis, EGCG inhibited VEGF-induced tube formation through the suppression of vascular endothelial-cadherin phosphorylation and AKT activation [146]. In normal human keratinocytes (NHK) that were stimulated by TNFα, EGCG at different concentrations (0.1 to 10 μM) decreased VEGF and interleukin-8 (proinflammatory cytokine involved in skin inflammation) [147]. Similarly, in human umbilical vein endothelial cells (HUVEC), EGCG inhibited VEGF in a concentration dependent manner and reduced cell proliferation [148]. On the same line, two works from the same research group used HUVEC cells (and MDA-MB231 breast cancer cell) and found that EGCG decreased VEGF production in a concentration-dependent manner (6.25, 25, 100 µM) [149,150]. They also demonstrated that, in HUVEC and MDA-MB231 mouse xenografts, in addition to cell line, green tea extract reduced the proliferation, tumor size, and tumor vessel density [150]. Mice fed with green tea extract in water *ad libitum* with a concentration of 0.62, 1.25, and 2.5 g/L (in which EGCG represented 46.8% weight). The effects were both time and concetration dependent. In human vascular smooth muscle cells (hVSMCs), EGCG was investigated after cell stimulation with platelet-derived growth factor (PDGF), which is known to induce VEGF expression [151]. EGCG blocked dose-dependently VEGF expression and PDGF receptor activation, in addition to ERK1/2 and AP-1. Additionally, in livestock ovarian cell model (swine granulosa cells), EGCG was evaluated as a food supplement. The catechin reduced VEGF production independently of oxygen condition and decreased cell growth, which suggested a role in angiogenic process, fundamental for follicle development [152].

### 3.9. Proteasome

The proteasome is a protein complex that is dedicated to degrade damaged or useless proteins that are unusable to cell [153]. It is strictly dependent on ubiquitins, molecules that tag unwanted proteins and continues with the degradation process mediated by proteases [154]. The proteasome has been investigated as a key target for multiple diseases, given its importance in cell physiology. Effectively, three approved drugs are available, bortezomib (for multiple myeloma and mantle cell lymphoma), carfilzomib and ixazomib (for relapsed or refractory multiple myeloma), and many others are in development [155]. Different cell types were tested with EGCG: human Jurkat T, prostate cancer (LNCaP, PC-3), breast cancer (MCF-7), normal (WI-38), and SV40-transformed (VA-13) human fibroblast cells [156,157]. The works showed that EGCG specifically blocked proteasome activity in LNCaP, PC-3, and MCF-7 cell extracts with a range of IC_50_ from 86 to 194 nm. In addition, in cell models, EGCG 1–10 µM induced the accumulation of two natural proteasome substrates, p27(Kip1) and IkappaB-alpha, an inhibitor of transcription factor NF-κB, together with G1 cell cycle phase arrest. Moreover, EGCG derivatives were studied both to understand the biochemical role of such molecules in proteasome inhibition and to find novel potent or stabilized alternatives to EGCG [158]. For example, a work of Kazi and collaborators revealed that the A-ring and gallate ester/amide bond are necessary for the blockade of proteasome [159]. Similarly, EGCG modified with peracetate presented higher proteasome-inhibitory activity if compared to EGCG, with increased cell death mechanism in Jurkat T cells [160]. Again, other EGCG analogs were tested and showed that B-ring/D-ring peracetate-protected EGCG derivatives could induce cell death [161,162]. Additionally, peracetate-protected EGCG in breast carcinoma cell line (MDA-MB-231) and in MDA-MB-231 mouse xenografts (daily *sub cute* injection with 50 mg/kg) was tested, showing tumor growth decrement, with concomitant proteasome inhibition and apoptosis activation [163]. Later, different fluoro-substituted EGCG analogs were generated and, among these, Pro-F-EGCG4 ((-)-(2R,3R)-5,7-Diacetoxy-2-(3,4,5-triacetoxyphenyl)chroman-3-yl3,4-difluorobenzoate) had potent effects than peracetate EGCG inducing apoptosis and decreasing cell proliferation (concentration-dependent effects from 10 to 50 µM) in the Jurkat and MDA-MB-231 cell lines, in addition to reducing tumor size in MDA-MB-231 mouse xenografts (daily *subcutaneous* injection with 50 mg/kg) [164,165]. In the same cell line (MDA-MB-231), EGCG was tested with catechol-σ-methyltransferase (COMT), an human enzyme that degrades catecholamines [166]. COMT can methylate EGCG, reducing its biological activity. The work reported that decreasing COMT activity could augment EGCG efficacy through apoptosis induction and proteasome inhibition. In YT (human natural killer) and Jurkat cell lines, EGCG (and other proteasome inhibitors) produced cell cycle modulation (subG0/G1 phase increase in Jurkat cells and G2/M phase arrest in YT cells), apoptosis with caspase-3 activation, and with mitochondrial membrane potential decrement [167]. Moreover, in Caco-2 cells, EGCG inhibited 20S proteasomes activity with antioxidant and proteasome blocking properties in cell lysates [168]. In the human neuroblastoma cell line (SH-SY5Y), EGCG at low concentration (1 µM) impacted on BCL2 associated agonist of cell death (Bad) levels, while the protein kinase C (PKC) inhibitor GF109203X impeded Bad degradation [169]. The authors claimed that EGCG promoted neuronal survival through the rapid removal of Bad (PKC-mediated) by proteasome, while using low doses, whereas most of works showed general EGCG inhibitory effects at 50 µM. This difference could probably reside in the specific cell model used. In a mouse model of experimental autoimmune encephalomyelitis (EAE), EGCG administered by gavage 300 μL per mouse twice daily was able to decrease clinical severity, as suggested by reduced brain inflammation and neuronal damage [170]. In addition, in the same work with human myelin-specific CD4+ T cells, EGCG impacted cyclin-dependent kinase 4 expression (with cell cycle arrest) and on 20S/26S proteasome complex activity (with subsequent NF-κB inhibition), demonstrating anti-inflammatory and neuronal protective capacities. In skin cancer cells (SCC-13 and A431), EGCG alone or associated to 3-deazaneplanocin A (a potent inhibitor of S-adenosylhomocysteine hydrolase, enzyme that is involved in homocysteine metabolism) suppressed cell cycle progression and increased apoptosis, through proteasome-dependent degradation of the polycomb group proteins, molecules that are involved in development, differentiation, and survival [171]. A study investigated the effects of extremely low frequency electromagnetic fields (ELFEF) on 20S proteasome while using EGCG in Caco-2 cells [172]. It is known that ELFEF can modulate intracellular ROS levels and the cell cycle progression, indeed EGCG was able to reduce the pro-oxidant effects, regulate cell cycle, and proteasome functionality. However, EGCG did not alter cell viability as measured by MTT assay.

Though numerous works demonstrated a positive interaction of EGCG in proteasome function, others showed a deleterious activity of EGCG. For example, in HepG2 cells, EGCG 50 µM exerted anti-cholesterolemic activity, reducing apolipoprotein B-100 (apoB) production and lipid assembly, thorough a proteasome-independent pathway, as demonstrated by a lack of response to N-acetyl-leucyl-leucyl-norleucinal, a proteasome inhibitor [173]. Moreover, the association of EGCG and bortezomib (proteasome inhibitor for multiple myeloma) should not be suggested, as their experimentation in human multiple myeloma cells (RPMI/8226 and U266) showed strong antagonism, with EGCG blocking proteasome inhibition being produced by bortezomib and, thus, stopping tumor cell death [174,175]. Similarly, in prostate cancer cells (PC3), bortezomib anti-tumor effects was counteracted by EGCG which induced autophagy while decreasing endoplasmic reticulum stress [176]. These data underline in a different cell model, as EGCG can antagonize bortezomib, blocking prostate cancer cells death, and thus negatively supporting tumor pro-survival.

### 3.10. Senescence

Senescence is defined the condition or process of deterioration with age [177]. Senescence plays a role in multiple biological events, such as embryogenesis, tissue regeneration and repair, ageing, cardiovascular, renal, liver diseases, and cancer [178,179]. EGCG has been implicated in senescence in different works. For example human mesenchymal stem cells (hMSCs) show the peculiar behavior of producing abundant ROS during culturing and this issue has been exploited with a natural anti-oxidant as EGCG [180]. The work explored the anti-senescent effect of EGCG in H_2_O_2_-exposed hMSCs and proved that EGCG reversed oxidative stress by downregulating the p53–p21 signaling pathway and upregulating Nrf2 expression. Similarly, in primary cells, including rat vascular smooth muscle cells (RVSMCs), human dermal fibroblasts (HDFs) and human articular chondrocytes (HACs), the same concentrations of EGCG (50 or 100 μM) could prevent senescence and recover cell cycle progression, again impacting on the p53 pathway at least in HDFs cells [181]. Furthermore, in 3T3-L1 preadipocytes, EGCG could inhibit stress-induced senescence (H_2_O_2_-induced) by countering DNA damage and cell cycle arrest [182]. In this work, EGCG (same concentrations as above, 50 or 100 μM) modulated PI3K/Akt/mTOR and AMPK pathways by blocking ROS, iNOS, Cox-2, NF-kB, and p53, and increasing apoptosis by suppressing antiapoptotic protein Bcl-2. Moreover, in WI-38 fibroblasts at late-stage (population doubling > 25) used as model of oxidative stress and inflammation, EGCG (25, 50, 100 µM) reduced TNF-α, IL-6, ROS, again acting on p53 [183]. ECGC also decreased retinoblastoma expressions, while enhancing E2F2 expressions and superoxide dismutase (SOD) 1 and 2, all being implicated in oxidation processes. EGCG has been shown to affect the development and severity of cellular senescence, but from a therapeutic perspective more work is needed. Senolytic attributes of EGCG are interesting, especially for attenuating age-associated disorders, though clinical trails and health consequence still need to be proposed and discovered.

### 3.11. Epigenetics and Mirnas

Epigenetics refers to heritable phenotype changes, in which DNA sequence alteration are not included. The most known epigenetic mechanisms involve DNA methylation and histone modification [184]. Extensively, epigenetics concerns all events “outside” DNA sequence modification; indeed, this paragraph wants to describe those mechanisms characteristic of DNA methylation and histone modification involving EGCG. Numerous works have been reported the significant impact of EGCG in epigenetic mechanism. For example, in 2003, EGCG was first studied as an inhibitor of DNA hypermethylation of CpG islands by acting on 5-cytosine DNA methyltransferase (DNMT). EGCG was used at 20 and 50 µM with concentration-dependent effects in human esophageal cancer cells (KYSE 510), colon cancer HT-29, and prostate cancer PC3 cells [185]. The authors showed that EGCG could block DMNT and reactivate methylated-silenced genes, as p16(INK4a), retinoic acid receptor β (RARβ), O-(6)-methylguanine methyltransferase (MGMT), and human mutL homologue 1 (hMLH1). Similar results were obtained in human breast cancer cell lines (MCF-7 and MDA-MB-231), whereas EGCG was able to block prokaryotic SssI DNA methyltransferase (DNMT) and human DNMT1 [186]. A recent work used the same cell lines (MCF-7 and MDA-MB-231) investigating RARβ expression and methylation, as mentioned in the first paragraph [52]. EGCG combined with 2-chloro-2’-fluoro-2’-deoxyarabinosyladenine (a deoxyadenosine analogue) reduced RARβ methylation with consequent increase of its expression, together with PTEN and CDKN1A, and with apoptosis induction and cell proliferation inhibition. EGCG acted on inhibitor of matrix metalloproteinases (TIMPs) and MMPs, being highly involved in tumorigenesis, in human prostate cancer cells (DUPRO and LNCaP) [187]. The authors showed that EGCG restored TIMP-3 expression and concomitantly increased histone H3K9/18 acetylation and decreased class I histone deacetylase, impacting on cell invasion and migration. Interestingly in serum of patients undergoing prostatectomy consuming 800 mg EGCG up to 6 weeks (participating to clinical trial NCT01340599), TIMP-3 levels were increased, suggesting a potential mechanism of action of EGCG in prostate cancer [187]. However, the trial failed to show a significant benefit of EGCG-treated prostate cancer patients [188].

Moreover, in a SKH-1 hairless mouse model, EGCG added to a hydrophilic cream and applied to approximately 1 mg/cm^2^ skin area could protect against photocarcinogenesis through the reduction of tumor incidence, tumor multiplicity, and tumor size [189]. Mechanisms at the basis of these results are related to the inhibition of DNA hypomethylation induced by ultraviolet B treatment. Likewise, in human skin cancer cells (A431) and squamous cell carcinoma (SCC 13) cells, EGCG could reactivate p16INK4a and Cip1/p21, impacting not only on DNA methylation and DNMT activity, but also on histone deacetylase (HDAC) activity and acetylated histones [190]. Again, in CD4+ Jurkat T cells, EGCG 50 µM could inhibit DNMT and induce a concomitant Foxp3 (a factor regulating the development and function of regulatory T cells) and IL-10 expression. Moreover, in mice that were treated with EGCG (injected intraperitoneally daily per mouse 50 mg/kg), there was an increase of regulatory T cells frequencies and numbers [191]. In human malignant lymphoma cells (CA46), EGCG alone (6 µg/mL) or combined with trichostatin A (inhibitor of HDACs) diminished p16INK4a gene methylation (thus increasing its expression) and provoked a cell proliferation decrement [192]. The effects were more evident with the combination of the compounds, rather than EGCG alone, being suggestive an additive or synergist effect. In human breast tumor tissues, expression of DNMTs was found increased by Mirza et al., consequently MCF-7 and MDA MB 231 cells were used as models for testing EGCG and other compounds [193]. The authors showed that EGCG decreased DNMT1, HDAC1, and methyl-CpG binding protein 2 (MeCP2 essential for nerve function), with a reversal of epigenetic changes present in cancer cells. On the same line, another work investigated a co-treatment of 5-aza-2’-deoxycytidine and EGCG in breast cancer cells (MCF-10A) and confirmed the effects on DNA methylation and histone modifications [194]. In a mouse lung tumor model that was induced by the tobacco-specific carcinogen 4-(Methylnitrosamino)-1-(3-pyridyl)-1-butanone (NNK), EGCG (0.5% in diet *ad libitum*) was implicated in DMNT1 reduction with simultaneous decrease of p-AKT, phospho-histone H2AX and tumor growth, again emphasizing the epigenetic effects of the catechin, potentially due the antioxidant activity of EGCG, as suggested by the authors. [195]. Furthermore EGCG was studied in the methylation status of the reversion-inducing cysteine-rich protein with Kazal motifs (RECK) gene, a tumour suppressor gene downregulating matrix metalloproteinases and blocking metastatic potential [196]. It has been shown in a salivary adenoid cystic carcinoma cells (SACC83) that the catechin was able to demethylate RACK gene and thus impact on invasion, migration, and metastasis. EGCG has been also implicated in autophagic process, as seen in Paragraph 3.1. It is known that in macrophages from aged mice Atg5 and LC3B genes are normally hypermethylated (genes regulating autophagy). The use of EGCG (probably a high dose, even if the authors did not report any cell viability change) in macrophages restored these genes expressions both in vivo and in vitro [197]. Similarly, in macrophages cells of cystic fibrosis (CF) mice, EGCG (administered intratracheally for three days with 25 mg/kg) restored Atg12 gene expression, improving autophagy in such a disease, where autophagic process highly impaired [198]. EGCG (10 *μ*M given diabetic pregnant mice at embryonic day 5.5 in drinking water) inhibited DNA hypermethylation in maternal diabetes-induced neural tube defects, decreasing the congenital defects and opposing the high glucose levels in embryo [199]. In another animal models, C57BL/6J male mice thatwere fed with high-fat diet (HFD) or control diet (CD), EGCG administered *ad libitum* at 25 mg/kg body weight per day in water modulated DNMT1 and MutL homologue 1 genes methylation, ameliorating inflammatory status in HFD mice [200]. In addition, EGCG enhanced gut microbiota, as evidenced by DNA damage in the *Firmicutes/Bacteroidetes* ratio and the total number of bacteria in the same work. In HeLa cells, EGCG abrogated DNMT3B and HDAC1 activity, while only the expression of DNMT3B was reduced, also affecting expression of retinoic acid receptor-β, cadherin 1, and death-associated protein kinase-1 [201]. A genome-wide study of DNA methylation reported the effects of EGCG in oral squamous cell carcinoma (CAL-27) [202]. The results showed that the main differentially expressed genes that were derived from MAPK, Wnt, and cell cycle signaling pathways, cell processes that are strongly associated cancer proliferation.

MicroRNA (miRNA) is a small non-coding RNA molecule that contains about 22 nucleotides that can post-transcriptionally regulate gene expression [203]. miRNAs are involved in numerous cell processes, including growth, proliferation, apoptosis, autophagy, metabolism, etc., and play a fundamental part in cancer. In prostate cancer cells (PC3) and PC3 mouse xenografts, EGCG blocked nuclear translocation and protein expression of androgen receptor (implicated in prostate cancer etiology), and these effects were associated with the down-regulation of androgen-regulated miRNA-21 and up-regulation of miRNA-330, a tumor suppressor [204]. Another target of EGCG was miRNA-210, being upregulated by the catechin in lung cancer cell lines (CL13, H1299, H460, A549), with induced accumulation of HIF-1α, which is a known miRNA-210 activator [205]. Moreover, in tobacco carcinogen-induced lung tumors in A/J mice, miRNAs profile was analysed after EGCG treatment (mice fed with purified diet containing 0.4% EGCG) [206]. It has been revealed that these miRNAs were strictly connected with MAPK, AKT, NF-κB pathways, and cell cycle regulation. Rat hepatoma cell line (FAO) were used to test different extracts and polyphenols, including EGCG [207]. The catechin reduced miR-33a and miR-122 levels (both being involved in dyslipidemia, cholesterol homeostasis, insulin resistance), with direct binding to miRNAs, as demonstrated by ^1^H NMR spectroscopy and, thus, could potentially impact cell metabolism. In non-small cell lung cancer cells (A549) mouse xenografts, EGCG that was associated with cisplatin was able to reduce tumor size, while in NCI-H460 cells (another non-small cell lung cancer cell line) was not [208]. This difference was imputable to different expression of miRNA-98-5p and miRNA-125a-3p after EGCG treatment, consequently, the authors suggested that EGCG and cisplatin could have a therapeutic role in some subtype of non-small cell lung cancer and could conveniently be associated. In nasopharyngeal carcinoma cell line (CNE2) differential miRNAs expression was observed, in particular 16 miRNAs with 20 µM EGCG-treated and 32 miRNAs with 40 µM EGCG-treated were modulated (>2-fold expression changes when compared with the control) [209]. Furthermore, while using an in vitro subarachnoid hemorrhage model (oxyhemoglobin added to PC12 cells), EGCG at 1 and 50 µM altered miRNAs expression and mainly impacted on the MAPK pathway (p38, Ca^2+^, autophagic activation), suggesting that the differential expression of miRNAs could be at the basis of the therapeutic efficacy of different concentrations of EGCG [210]. EGCG in a dose- and time-dependent manner inhibited the proliferation and regulated miRNAs expression of miR-210, miR-29a, miR-203, and miR-125b in different cervical carcinoma cells in multiple human cervical cancer cell lines (HeLa (HPV16/18+), SiHa (HPV16+), CaSki (HPV16+), and C33A (HPV-)) [211]. In osteosarcoma cell lines (MG-63 and U-2OS), EGCG induced miRNA-1 upregulation and inhibited c-MET expression, while its combination with c-MET inhibitor enhanced inhibitory effects on tumor cell growth, heading to cell proliferation reduction, cell cycle arrest, and apoptosis induction [212]. EGCG downregulated Ca^2+^ (CRAC) channel (Orai1), its regulator stromal cell-interaction molecule 2 (STIM2), and store operated Ca^2+^ entry (SOCE), all being implicated in Ca^2+^ regulation, in murine CD4+ T cells and human Jurkat T cells. The work showed that EGCG repressed Ca^2+^ entry by up-regulation of miR-15b, which ruled STIM2, Orai1, SOCE, and acted on AKT/PTEN pathway [213]. In anaplastic thyroid carcinoma cells (SW1736 and 8505C), EGCG and other natural compounds had heterogeneous effects, with curcumin being the most effective [214]. Nonetheless, EGCG provoked a significant reduction of miRNA-221 only, even if five important miRNAs that were implicated in thyroid cancer progression were studied (miRNA-221, miRNA-222, miRNA-21, miRNA-146b, and miRNA-204). In multiple myeloma cells (MM1.s), EGCG (1 and 5 μM) was shown to inhibit miRNA-25, miRNA-92, miRNA-141, and miRNA-200a, all being implicated in p53 targeting reduction, a hallmark of multiple myeloma [215]. The suppression of these miRNAs could have a potential effect in the treatment of such disease, as the authors claimed that the EGCG doses were physiologically relevant and within the range of human consumption. In pancreatic ductal adenocarcinoma cell lines (BxPc-3, MIA-PaCa2, and CRL-1097), EGCG and other catechins (also associated with quercetin and sulforaphane) were used and showed to reduce apoptosis, migration, and the matrix metalloproteinases MMP-9 and MMP-2, with the activation of miR-let-7 leading to the inhibition of its target gene K-Ras [216]. Of note that EGCG was not among the most effective compounds, whereas the combination of sulforaphane, quercetin, and catechins extract seemed to have a strong therapeutic action.

### 3.12. Other Mechanisms

The effects of EGCG have been also reported in many other signaling pathways and other diseases other than cancer could be involved in the benefit of the catechin. For example, EGCG treatment inhibited VEGF signaling, leading to a reduction in mitogenesis of human umbelical arterial cells [217]. Mitogenesis is key process for cell energy and its modulation can impact the survival, growth, and replication. As already reported in paragraphs 3.8 and 3.10, the inhibition of some components of MMP family by EGCG could reduce tumor invasion and metastasis. For instance, MMP-2 and MMP-9 are both inhibited by EGCG in the prostate of transgenic adenocarcinoma of mouse prostate (TRAMP) model, as mentioned before [89], as well in endothelial cells [218]. In addition, the anticancer properties of EGCG may be dependent on the inhibition of the protease urokinase-plasminogen activator (uPA), which is often overexpressed in human tumors and whose activity is involved in the regulation of apoptosis, cell replication, migration, and adhesion [219]. EGCG was also studied in the amyloid aggregation and neurotoxicity model. The catechin effects were examined in relation to misfolded or misassembled proteins in human islet amyloid polypeptide (hIAPP) transgenic mice, an animal model for investigating the formation of amyloid fibrils and, thus, pancreas dysfunction and diabetes [220]. EGCG was administered at 60 mg/kg in drinking water for three weeks and revealed potential benefit in reducing amyloid fibril formation. Another work explored the effect of EGCG on the aggregation process and toxicity of the expanded ataxin-3 (AT3), the polyglutamine (polyQ)-containing protein responsible for spinocerebellar ataxia type 3 (SCA3) [221]. EGCG interfered with fibril aggregation and blocked mature fibrils formation of both the cell and animal model. Indeed, in fibroblast-like cells (COS-7), co-incubation of EGCG and AT3 diminished the toxicity of protein aggregates, as also demonstrated in in vivo model (*Caenorhabditis elegans* simplified SCA3), whereas EGCG increased motility of affected worms. In Alzheimer disease, amyloid β (Aβ) accumulates in the brain, leading to neuronal cell death. EGCG reduced the inflammation and neurotoxicity in EOC 13.31 mouse cell line, with a mechanism suppressing TNFα, IL-1β, IL-6, iNOS, NF-κB, JNK, p38, ERK, and restoring Nrf2/HO-1 antioxidant signaling in neuroinflammatory responses of Aβ -stimulated microglia [222]. Moreover, the effect of EGCG on cAMP activated protein kinase (PKA) is currently under investigation: pioneeristic studies were considering a possible role of EGCG in the regulation of glucose homeostasis via PKA inhibition downregulation of FoxO1. The proposed model needs further confirmation, but opens a new door for EGCG [223].

## 4. Conclusions

EGCG is responsible for a wide variety of biological activities, including cancer prevention. The preclinical activities of EGCG are associated with proliferation, differentiation, apoptosis, inflammation, angiogenesis, and metastasis. More efforts are needed to translate the plethora of preclinical data in effective human therapeutic options.

## Figures and Tables

**Figure 1 molecules-25-00467-f001:**
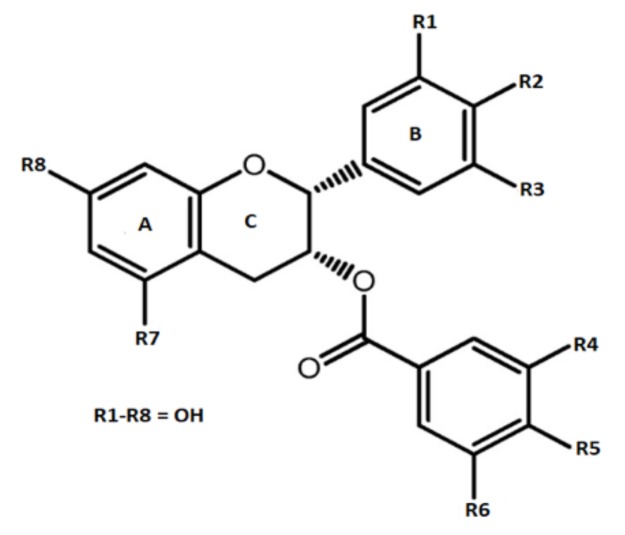
Chemical structure of epigallocatechin gallate.

**Figure 2 molecules-25-00467-f002:**
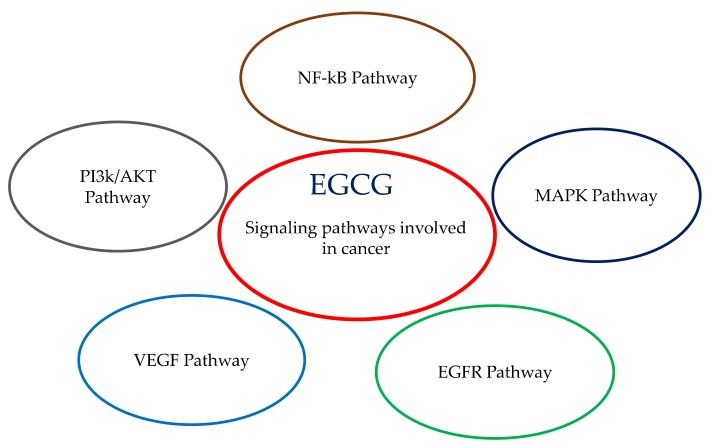
Epigallocatechin gallate (EGCG) involved in the signaling pathways in cancer.

**Table 1 molecules-25-00467-t001:** Occurrence of epigallocatechin-3-gallate in different food.

Description	Content	Reference
Japanese green tea	18.1–23.1 mg/g	[44]
Long-jing tea	32.9–35.5 mg/g	
Jasmine tea	29.8–31.0 mg/g	
Pu-erh tea	16.9–19.19.1 mg/g	
Iron Buddha tea	0.12–0.30 mg/g	
Oolong tea	11.8–12.2 mg/g	
Ceylon tea	7.4–8.9 mg/g	
Green tea	4.62 mg/100 mL	[45]
Black tea	1.35 mg/100 mL	
Ban-cha (tea leaves)	12.2–27.3 mg/g DM	[46]
Fukamushi-cha (tea leaves)	13.8–18.6 mg/g DM	
Yame-cha (tea leaves)	19.7–32.9 mg/g DM	
Uji-cha tea leaves)	22.2–35.9 mg/g DM	
Sayama-cha (tea leaves)	21.5–37.2 mg/g DM	
Uji-matt-cha (Powder tea)	20.1–29.6 mg/g DM	
Gyokuro-cha (Powder tea)	29.9–37.2 mg/g DM	
Sen-cha (tea bags)	12.9–23.6 mg/g DM	
Green tea (Bagged leave form)	54.3–153.0 mg/g dry tea	[47]
Green tea (Loose leave form)	56.5–205.0 mg/g dry tea	
White tea (Bagged leave form)	46.0–154.0 mg/g dry tea	
White tea (Loose leave form)	38.9–144.0 mg/g dry tea	
Green tea (Infusions)	117 to 442 mg/l	[48]
Typhoo,	7.9 mg/230 mL (mg/serving)	[49]
Tesco standard blend,	4.7 mg/230 mL (mg/serving)	
Tesco premium blend,	5.6 mg/230 mL (mg/serving)	
Sainsbury red blend,	8.5 mg/230 mL (mg/serving)	
Sainsbury gold blend,	11.8 mg/230 mL (mg/serving)	
PG Tips,	7.1 mg/230 mL (mg/serving)	
Tetley	5.7 mg/230 mL (mg/serving)	
Apples, Fuji, raw, with skin	1.93 mg/100 g edible portion	[50]
Apples, Golden Delicious, raw, with skin	0.19 mg/100 g edible portion	
Apples, Granny Smith, raw, with skin	0.24 mg/100 g edible portion	
Apples, Red Delicious, raw, without skin	0.46 mg/100 g edible portion	
Apples, Red Delicious, raw. with skin	0.13 mg/100 g edible portion	
Avocados, raw,	0.15 mg/100 g edible portion	
Blackberries, raw (Rubus spp.)	0.68 mg/100 g edible portion	
Cranberries, raw	0.97 mg/100 g edible portion	
Peaches, raw	0.30 mg/100 g edible portion	
Pears, raw	0.17 mg/100 g edible portion	
Plums, black diamond, with peel, raw	0.48 mg/100 g edible portion	
Raspberries, raw	0.54 mg/100 g edible portion	
Nuts, hazelnuts or filberts	1.06 mg/100 g edible portion	
Nuts, pecans	2.30 mg/100 g edible portion	
Nuts, pistachio nuts, raw	0.40 mg/100 g edible portion	
Tea, black, brewed, prepared with tap water	9.36 mg/100 g edible portion	
Tea, black, brewed, prepared with tap water, decaffeinated	1.01 mg/100 g edible portion	
Tea, fruit, dry	415.0 mg/100 g edible portion	
Tea, green, brewed	64.0 mg/100 g edible portion	
Tea, green, brewed, decaffeinated	26.0 mg/100 g edible portion	
Tea, green, large leaf, Quingmao, dry leaves	7380 mg/100 g edible portion	
Tea, oolong, brewed	34.48 mg/100 g edible portion	
Tea, white, brewed	46.0 mg/100 g edible portion	
Tea, white, dry leaves	4245 mg/100 g edible portion	
Carob flour	109.46 mg/100 g edible portion	

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
