# Peer review of "Preclinical Activities of Epigallocatechin Gallate in Signaling Pathways in Cancer"

_molecules, 2020, doi:10.3390/molecules25030467_

Round 1

Reviewer 1 Report

    The anticancer mechanism of tea EGCG was partially reviewed in the manuscript. The topic is interesting and can be published after major revision.

    1.There have been several review papers on this topic. Why this review is necessary to be published was not clarified in the section of INTRODUCTION.

    2.The methods how to collect and screen the literatures should be described.

    3.The descriptions without relating to the present topic should be deleted. For example, EGCG plays important role in treating cardiovascular and metabolic diseases [37] in Line 109.

    4.Authors’ contributions should be included.

    5.The relationship of section 2 including Table 1 to the topic of this review was not clarified.

    6.The protective effects of EGCG against cancers include many aspects. The main content of this review was focused on the “Signaling Pathways Involving EGCG”. I suggest the author to revise the title accordingly.

    7.Pure EGCG is white in colour, except for the oxidized one. The description in Lines 86-87 should be revised accordingly: Pure epigallocatechin gallate is classified as an odourless crystal and/or powder available in white, pink or cream colour.

    8.The expression errors should be checked, such as that in Line 28: “Epigallocatechin gallate (EGCG) is the main bioactive catechin predominantly present in…” should be replaced by “Epigallocatechin gallate (EGCG) is a main bioactive component of catechins predominantly present in…”

    9.The format of the references does not meet the requirements of the MOLECULES.

Author Response

We thank reviewer comments for their help and suggestions necessary for the improvement of the manuscript. For convenience we highlighted in yellow any change made in the manuscript.

Comments and Suggestions for Authors
The anticancer mechanism of tea EGCG was partially reviewed in the manuscript. The topic is interesting and can be published after major revision.
1.There have been several review papers on this topic. Why this review is necessary to be published was not clarified in the section of INTRODUCTION.
Answer:
Thank you so much for your comments. This review article having a review focused on EGCG, cancer and signaling pathways.
2.The methods how to collect and screen the literatures should be described.
Answer
A new paragraph was added to the manuscript:
“This review was prepared searching for articles, papers, books from different sources, such as Embase-Elsevier, Google Scholar, Ovid, PubMed, Science Direct, Scopus, Web of Science. The search strategy was based on the combination of different keywords, such as EGCG, cancer, tumor, mechanism of action, preclinical, clinical data, considering any time of publication. Only sources written in English from any country were included.”
3.The descriptions without relating to the present topic should be deleted. For example, EGCG plays important role in treating cardiovascular and metabolic diseases [37] in Line 109.
Answer
We deleted out of topics sentence and reference. (EGCG plays important role in treating cardiovascular and metabolic diseases [37]).

4.Authors’ contributions should be included.
Answer
Authors contribution is at the end of the manuscript, after “Abbreviations”.

5.The relationship of section 2 including Table 1 to the topic of this review was not clarified.
Answer
We added a new sentence at the beginning of section 2:
This section wants to underline the importance of EGCG in foods and preparations for human dietary. It is well known the close relationship between food and disease (Ref), thus EGCG consumed in food could be a natural potential source of health benefits, even in complex and intricated disorder as cancer.
6.The protective effects of EGCG against cancers include many aspects. The main content of this review was focused on the “Signaling Pathways Involving EGCG”. I suggest the author to revise the title accordingly.
Answer
We followed the suggestion of the reviewer and modified the title. Now it is: “Preclinical Pharmacological Activities of Epigallocatechin-3-gallate in Signaling Pathways: An Update On Cancer”.
7.Pure EGCG is white in colour, except for the oxidized one. The description in Lines 86-87 should be revised accordingly: Pure epigallocatechin gallate is classified as an odourless crystal and/or powder available in white, pink or cream colour.
Answer
The edits were made.
8.The expression errors should be checked, such as that in Line 28: “Epigallocatechin gallate (EGCG) is the main bioactive catechin predominantly present in…” should be replaced by “Epigallocatechin gallate (EGCG) is a main bioactive component of catechins predominantly present in…”
Answer
We corrected the error in the Abstract.
9.The format of the references does not meet the requirements of the MOLECULES.
Answer
We corrected and revised all the references.

Reviewer 2 Report

The authors analyzed the pharmacological activities of 2-Epigallocatechin-3-gallate.

The paper is well written and is of interest but there are some issues that have to be addressed before considering it suitable for publication.

Authors evaluated the effect of this molecules on apoptosis, necrosis, autophagy and on their related pathways. They overlooked to evaluate effects of epigallocatechin on senescence. This phenomenon is strictly related to cell cycle arrest, reactive oxygen species and autophagy. Please, see for example

Int J Mol Med. 2016 Oct;38(4):1075-82. doi: 10.3892/ijmm.2016.2694. Epigallocatechin-3-gallate prevents oxidative stress-induced cellular senescence in human mesenchymal stem cells via Nrf2. Oxidative Medicine and Cellular Longevity Volume 2012 http://dx.doi.org/10.1155/2012/850684

Preventive Effects of Epigallocatechin-3-O-Gallate against Replicative Senescence Associated with p53 Acetylation in Human Dermal Fibroblasts

 Biogerontology 20(16) · November 2018  Epigallocatechin gallate suppresses premature senescence of preadipocytes by inhibition of PI3K/Akt/mTOR pathway and induces senescent cell death by regulation of Bax/Bcl-2 pathway   DOI: 10.1007/s10522-018-9785-1 DOI: 10.1039/C9RA03313K (Paper) RSC Adv., 2019, 9, 26787-26798 Epigallocatechin gallate prevents senescence by alleviating oxidative stress and inflammation in WI-38 human embryonic fibroblasts For general information about senescence, autophagy and proteasome you may read

Oncotarget. 2015 Nov 24;6(37):39457-68. doi: 10.18632/oncotarget.6277.

PMID: 26540573

Author Response

Comments and Suggestions for Authors
The authors analyzed the pharmacological activities of 2-Epigallocatechin-3-gallate.

The paper is well written and is of interest but there are some issues that have to be addressed before considering it suitable for publication.
Authors evaluated the effect of this molecules on apoptosis, necrosis, autophagy and on their related pathways. They overlooked to evaluate effects of epigallocatechin on senescence. This phenomenon is strictly related to cell cycle arrest, reactive oxygen species and autophagy. Please, see for example
Int J Mol Med. 2016 Oct;38(4):1075-82. doi: 10.3892/ijmm.2016.2694. Epigallocatechin-3-gallate prevents oxidative stress-induced cellular senescence in human mesenchymal stem cells via Nrf2. Oxidative Medicine and Cellular Longevity Volume 2012 http://dx.doi.org/10.1155/2012/850684
Preventive Effects of Epigallocatechin-3-O-Gallate against Replicative Senescence Associated with p53 Acetylation in Human Dermal Fibroblasts
 Biogerontology 20(16) · November 2018  Epigallocatechin gallate suppresses premature senescence of preadipocytes by inhibition of PI3K/Akt/mTOR pathway and induces senescent cell death by regulation of Bax/Bcl-2 pathway   DOI: 10.1007/s10522-018-9785-1 DOI: 10.1039/C9RA03313K (Paper) RSC Adv., 2019, 9, 26787-26798 Epigallocatechin gallate prevents senescence by alleviating oxidative stress and inflammation in WI-38 human embryonic fibroblasts For general information about senescence, autophagy and proteasome you may read
Oncotarget. 2015 Nov 24;6(37):39457-68. doi: 10.18632/oncotarget.6277.
PMID: 26540573
Answer
We added a new paragraph (3.11 Senescence) with the suggestions of the reviewer

Reviewer 3 Report

The manuscript provide a comprehensively discussion on the biological activity of EGCG.  In general, the manuscript is well organized, but several points should be revived further.

(1) In order to improve the content to be readable, the authors should draw the text’s content in Figures.

(2) Given that EGCG is water soluble, the authors should discuss the mechanism of EGCG internalization or the binding of EGCG with cell surface receptor(s) for displaying its biological activities.

(3) The authors should discuss the mechanism responsible for the multiple activities of EGCG in different cell-type.

(4) Line 515, “stabilized” should be “stabilized”; Line 700, “ inveolved” should be “ involved”.

Author Response

Comments and Suggestions for Authors
The manuscript provide a comprehensively discussion on the biological activity of EGCG. In general, the manuscript is well organized, but several points should be revived further.
(1) In order to improve the content to be readable, the authors should draw the text’s content in Figures.
Answer
We added a new figure with signaling pathways involved in cancer (Figure 2).
(2) Given that EGCG is water soluble, the authors should discuss the mechanism of EGCG internalization or the binding of EGCG with cell surface receptor(s) for displaying its biological activities.
Answer
(3) The authors should discuss the mechanism responsible for the multiple activities of EGCG in different cell-type.
Answer
We added a new sentence discussing this issue at the beginning of section 3.
EGCG possesses multiple pharmacological activities, depending on cancer cell type, thus reflecting different protein expression and epigenetic mechanisms. Every cell model can express and modulate at convenience its genetic background and this is at the basis of the diverse biomolecular targets triggered by EGCG.
(4) Line 515, “stabilyzed” should be “stabilized”; Line 700, “ inveolved” should be “ involved”.
Answer
The above errors were corrected.

Round 2

Reviewer 1 Report

All the comments were addressed and I recommend its publication in Molecules.